# Towards a New Analytical Creep Model for Cement-Based Concrete Using Design Standards Approach

**Pablo Peña Torres, Elhem Ghorbel *** and **George Wardeh**

Laboratory of Mechanics and Materials of Civil Engineering (L2MGC), CY Cergy Paris University, 5 Mail Gay LUSSAC, Neuville-sur-Oise, 95031 Cergy-Pontoise CEDEX, France; ppenat@uni.pe (P.P.T.); george.wardeh@cyu.fr (G.W.)
* Correspondence: elhem.ghorbel@cyu.fr

**Abstract:** Creep properties are determined in design standards by measuring the creep coefficient, noted $\varphi$, as a function of time, $t$, and the age of the concrete at loading, $t_0$. The work aims to study the validity of the analytical models proposed in the most used international standards and to check the possibility of their extension to estimate the creep of recycled aggregates concrete (RAC). A database was built from experimental results available in bibliographic references including 121 creep curves divided into 73 curves for natural aggregates concrete (NAC) and 48 curves for RAC. The comparison between the experimental and predicted values showed a significant dispersion for NAC and RAC. For the remediation of this dispersion, a new analytical model was developed for NAC. The parameters being the conventional creep coefficient, $\varphi_0$, the power of the ageing function, named $\alpha$, and $\beta_h$, which accounts for the relative humidity and the compressive strength in the ageing function, were identified by inverse analysis. It was found that the power of the ageing function is 0.44 and not 0.3, as fixed by Eurocode 2 (EC2). Moreover, new expressions were proposed for $\varphi_0$ and $\beta_h$. The presence of recycled aggregates was considered through the equivalent replacement ratio.

**Keywords:** concrete; recycled aggregates; creep; design standards; analytical models

## 1. Introduction

Creep represents the ability of concrete to deform under constant sustained load. It has a considerable effect on the performance of concrete elements because it leads to a volume change, an increase of beams and slabs deflection, as well as a stress redistribution [1–4]. Long-term behavior depends on several parameters, such as the age of concrete at the moment of loading [5], the curing and environmental conditions [5], the stress level [6], the duration and the rate of loading [6], the concrete member size [6], as well as the material composition [6,7]. Moreover, studies relative to concrete incorporating recycled aggregates (RAC) showed that creep deformation is higher than those of natural aggregates concrete (NAC) [8–10]. This increase in creep deformations is mainly due to the presence of the attached mortar and the phenomenon is amplified if recycled fines are used [8,11,12].

Domingo-Cabo et al. [11] studied the influence of replacement ratio on the creep when the formulation parameters are kept constant. The researchers showed that the basic creep of RAC, incorporating 20% of recycled coarse aggregate, is 25% higher than that of the NAC of control. For a replacement ratio of 50%, the increase in creep strain was 29%, while for RAC with 100% recycled concrete, the increase was 32%. In addition, the increase in the total creep strain was more pronounced. The authors finally concluded that the creep models proposed in design standards conservatively predict the creep strain of RAC.

Fathifazl et al. [8] were interested in the influence of mix formulation parameters on creep strains. They showed that with a mortar content (attached and new) equal to that of the control concrete, the creep deformations are the same. On the other hand, by using a conventional mix design method consisting in replacing natural aggregates by recycled ones, the creep strains increase. For this latter series of concrete, the authors

concluded that analytical models proposed by ACI [13] and CEB-FIB [14] standards require modification by introducing a factor which accounts for the effect of residual mortar volume fraction in RAC. However, they did not measure this parameter experimentally, but they demonstrated its calculation based on the physical properties of recycled aggregates.

Knaack and Kurama [15] investigated the effect of the substitution percentage of natural aggregates by recycled ones and the influence of the recycled material source on the long-term behavior of concrete. In this study, three recycled coarse sources and two substitution ratios were used for concrete formulation, and other parameters such as cure conditions, loading age, and loading level were considered. The obtained results showed that recycled coarse aggregates increase the magnitude of creep strains and these latter are inversely proportional to the strength of the parent concrete. The authors also showed that the ACI model [13] could not be used for RAC without considering an adjustment factor calculated by linear regression.

Gholampour and Ozbakkaloglu [16] investigated the quality effect of recycled coarse aggregate (RCA) on the instantaneous and the long-term behavior of two sets of concrete. The first series is a normal strength concrete (NSC), while the second, called HSC, is a high-performance one. For each set, two RACs were derived from a reference concrete by using either a low-strength or high-strength parent for RCA. The obtained experimental results showed that the strength of the parent concrete plays a significant role in the creep behavior of RAC. Concrete incorporating low-strength RCA showed lower mechanical strength and higher creep strains than concrete mixed with high-strength RCA. Furthermore, it was found that HSC mixed with high-strength RCA has the same instantaneous and long-term performance as the reference NAC. The behavior difference can be explained by the fact that the quality of the attached interface is better when aggregates come from concrete with good mechanical strength.

Lye et al. [17] performed a bibliographic analysis on the creep of RAC taken form 27 countries between 1984 and 2015. They found that the average increase in creep of RAC with 100% coarse RCA is 32% higher than that of the corresponding NAC; while for a 20% coarse RCA content, the increase is 20%. Based on their analysis, they proposed a diagram for the creep prediction, which is not simply usable.

Other studies will be presented and analyzed thereafter, but the common point between all the works is that the authors have tried to model the creep behavior of RAC based on their results by admitting the validity of analytical models for NAC [9,18–23]. In addition, some authors propose modifications with factors that are often difficult to measure [8,9,23].

The first aim of the present paper is to revise the applicability of analytical models adopted in design standard for NAC and to study their possible extension to RAC using a new database built from the experimental results available in the literature. The second aim is to develop a new analytical expression for NAC which can be extended to RAC simply by considering the equivalent replacement ratio. To achieve the main objective, a new creep database including 121 curves divided into 73 curves for natural aggregates concrete (NAC) and 48 curves for recycled aggregates concrete (RAC) was built from experimental results available in bibliographic references. With the help of this database, the analytical expressions proposed by the most communally used design standards were verified for NAC and new analytical expressions have been proposed. Moreover, the modifications necessary to take into account the presence of recycled aggregates were introduced.

## 2. Concrete Creep

The strain of concrete increases with time under sustained stress due to creep. The strain occurring during the application of load is named the instantaneous strain, $\varepsilon e\,(t_0)$. The elastic strain depends mainly on the elastic modulus of concrete at age $t_0$. The total strain, instantaneous plus creep, is given by:

$$\varepsilon_{c\sigma}(t, t_0) = \sigma_c(t_0) J(t, t_0) \tag{1}$$

where: $J(t,t_0) = \frac{1+\varphi(t,t_0)}{E_{cm}(t_0)}$. $E_{cm}(t_0)$ is the elastic modulus at the moment of loading and $\varphi(t,t_0)$ is the dimensionless creep coefficient, which represents the ratio of creep to the instantaneous strain. Its value increases with the decrease of age at loading $t_0$ and the increase of the length of the loading period.

## 2.1. Creep Prediction Models

　　　　Three models were studied and compared in this study for the creep prediction. The models include ACI 209.2R-08 [13], EC2 [24], and fib Model Code 2010 [25]. The analytical expressions of the studied models are summarized in Table 1 and the required parameters for each model are given in Appendix A Table A1. The CEB-FIB90 model [14] was not considered since it is identical to the EC2 one. The basic assumption for EC2 and ACI models is that the creep coefficient, $\varphi(t,t_0)$, is the product of the ultimate creep coefficient by a time-dependent function, which tends towards one at the infinity. However, for fib Model Code 2010, the creep coefficient is the sum of two contributions, namely the basic creep, $\varphi_{bc}(t,t_0)$, and the drying creep, $\varphi_{dc}(t,t_0)$. It can be noticed from Appendix A Table A1 that ACI 209R-08 [13] requires the greatest number of parameters for creep prediction. EC2 [24] and fib MC2010 models [14,25] need the same number of parameters. For all studied models, the common parameters are the relative humidity, the age of concrete at the moment of loading, and the type of cement. The slump at fresh state, the percentage of fine aggregates by weight, and the air content are only used by ACI 209.2R-08 [13]. The specific parameters to EC2 and fib MC2010 models are the temperature, the type of cement, the stress level, the cross-sectional area, and the perimeter of the member in contact with the atmosphere.

**Table 1.** Studied creep prediction models.

| Model | Equation | Validity |
|---|---|---|
| EC2 * [24] | $\varphi(t,t_0) = \left[1 + \frac{1-\frac{RH}{100}}{0.1\sqrt[3]{h_0}}\right]\left[\frac{16.8}{\sqrt{f_{cm}}}\right]\left[\frac{1}{0.1+t_{0,adj}^{0.2}}\right]\left[\frac{t-t_0}{\beta_H+t-t_0}\right]^{0.3}$ | $f_{cm} \le 35$ MPa |
| | $\varphi(t,t_0) = \left[\left(1 + \frac{1-\frac{RH}{100}}{0.1\sqrt[3]{h_0}}\left[\frac{35}{f_{cm}}\right]^{0.7}\right)\left[\frac{35}{f_{cm}}\right]^{0.2}\right]\left[\frac{16.8}{\sqrt{f_{cm}}}\right]\left[\frac{1}{0.1+t_{0,adj}^{0.2}}\right]\left[\frac{t-t_0}{\beta_H+t-t_0}\right]^{0.3}$ | $f_{cm} > 35$ MPa |
| MC 2010 [25] | $\varphi(t,t_0) =$ $\left[\frac{1.8}{f_{cm}^{0.7}}\right]Ln\left(\left(\frac{30}{t_{0,adj}}+0.035\right)^2(t-t_0)+1\right) + \left[\frac{412}{f_{cm}^{1.4}}\right]\left[\frac{1-\frac{RH}{100}}{\sqrt[3]{0.1\frac{h_0}{100}}}\right]\left[\frac{1}{0.1+t_{0,adj}^{0.2}}\right]\left[\frac{t-t_0}{\beta_h+t-t_0}\right]^{\gamma(t_0)}$; $\gamma(t_0) = \frac{1}{2.3+\frac{3.5}{\sqrt{t_{0,adj}}}}$ | $\sigma_c(t_0) \le 0.4f_{cm}(t_0)$ |
| | $\varphi(t,t_0) =$ $\left(\left[\frac{1.8}{f_{cm}^{0.7}}\right]Ln\left(\left(\frac{30}{t_{0,adj}}+0.035\right)^2(t-t_0)+1\right) + \left[\frac{412}{f_{cm}^{1.4}}\right]\left[\frac{1-\frac{RH}{100}}{\sqrt[3]{0.1\frac{h_0}{100}}}\right]\left[\frac{1}{0.1+t_{0,adj}^{0.2}}\right]\left[\frac{t-t_0}{\beta_h+t-t_0}\right]^{\gamma(t_0)}\right) \times$ $exp\left[1.5\left(\frac{|\sigma_c|}{f_{cm}}-0.4\right)\right]$ | $0.4f_{cm}(t_0) < \sigma_c(t_0) \le 0.6f_{cm}(t_0)$ |
| ACI 209.2R–08 [13] | $\varphi(t,t_0) = 2.35\frac{(t-t_0)^{0.6}}{10+(t-t_0)^{0.6}}$ | Standard |
| | $\varphi(t,t_0) = 2.35\left(\gamma_{c,t_0}\cdot\gamma_{c,RH}\cdot\gamma_{c,VS}\cdot\gamma_{c,S}\cdot\gamma_{c,\Psi}\cdot\gamma_{c,\alpha}\right)\frac{(t-t_0)}{d+(t-t_0)}$; $\mathbf{c}$ | |

\* *For $\sigma_c(t_0) > 0.45f_{cm}(t_0)$, the value of the creep coefficient is $\varphi(t,t_0)\cdot exp[1,5\cdot(|\sigma_c|/f_{cm}(t_0)-0.45)]$.*

　　　　Appendix A Table A2 shows the standard conditions under which the models are applied. For conditions other than the standard conditions, the value of the ultimate creep coefficient needs to be modified by correction factors, as shown in Table 1. For the other models, it was found that the only correction necessary is when the loading level exceeds 50% of the compressive strength.

　　　　Many researchers have tried to modify the expressions available in the literature for creep prediction to take into account the presence of recycled aggregates [8,9,15,17–23]. The parameters adopted by each work for the proposed modifications are summarized in Appendix A Table A3. Fathifazl et al. [8] used the model of Neville [26,27], in which the creep coefficient of concrete is related to the creep coefficient of its cement paste

and to the volume fraction of its natural aggregate. The authors introduced a corrective term in order to take into account the presence of recycled aggregates in concrete mixes, called residual mortar coefficient. Fathifazl et al. [8] showed a good agreement between experimental and analytical results obtained only in their study. Fan et al. [19] also used the model of Neville [26,27] in order to model their experimental results. They proposed a modification to the initial expression of Neville by introducing the volume fraction of RCA and the properties of adhered mortar. However, the results have not been generalized for other values available in the literature. Fathifazl and Razaqpur [18] introduced a RCA modification coefficient in the analytical model of ACI 209.2R-8. The proposed modification improved the prediction of experimental results obtained by the authors, but it has not been verified with other results. Geng et al. [21,23] used the basic definition of creep coefficient (Equation (1)) and Neville's model and modified them to account for RCA. The proposed modification takes into account the water absorption and the density of RCA as well as the microstructural modifications related to RCA incorporation in concrete. However, the models proposed are limited to the authors' own results. Indeed, all the previous models assume the knowledge of the volume of attached mortar as well as knowledge of the physical and mechanical properties of recycled aggregates, which hinders their generalization. Silva et al. [20] introduced conservative correction factors to improve the creep coefficient predilection of RAC in the most widely used models. Knaack and Kurama [15] followed the same logic and introduced a correction factor of the ultimate creep expression proposed by the ACI 209-R. For these two studies, no modification was proposed for the aging function. The work of Lye et al. [17] is also part of the same framework, where a correction factor taking into account the percentage of RCA is introduced in the basic expressions of creep perdition models. More recently, Tošic et al. [9] adopted this approach to improve the fib MC2010 model by introducing a coefficient that takes into account both the compressive strength and the percentage of recycled aggregates.

Using the form of the CEB-FIB90 model, Pan and Meng [22] were almost the first who were interested not only in the ultimate creep coefficient but in the power of the time development function. The authors just gave new values based on their own results only.

### 2.2. Experimental Database

Many databases have been proposed in the bibliographical references, but unfortunately they do not provide all the necessary information for the requalification of creep analytical models [17,20,28]. For this reason, a database consisting of 73 NAC formulations and 48 RAC mixtures was established in this work based on the available studies in the literature [8,11,16,21,22,29–38]. The chosen references provide the following information:

- 28-day experimentally measured compressive strength and elastic modulus
- Concrete's age at loading and the applied stress level
- Concrete's mix proportion
- Type of cement and aggregate properties
- Dimensions of the test specimens
- Curing conditions and environmental conditions during creep test
- Creep strain curves after 90 days of loading

This database, despite the reduced number of experimental points compared to the other databases [9,20,28], considers not only the final creep coefficient but all the creep over time curves. For the acquisition of experimental curves, the free software PlotDigitizer (http://plotdigitizer.sourceforge.net, 2015) was used. Appendix A Tables A4 and A5 summarize the parameters provided by each selected study and the range of parameters' variation. The statistical distribution of these parameters is represented by Figure A1 given in the Appendix A.

For all the chosen mixtures, the equivalent substitution ratio was calculated according to the following expression [39]:

$$\Gamma_m = \frac{\sum M_{RA}}{\sum M_{(NA\ +\ RA)}}$$

(2)

where $M_{RA}$ and $M_{(NA+RA)}$ are the weight of recycled aggregates and the total weight of natural and recycled aggregates, respectively.

## 3. Result Analysis of Prediction Models for NAC

Figure 1 shows a comparison between creep coefficients calculated according to models presented in Table 1 and creep coefficients deduced from experimental creep-time curves after 90 days of loading. It is clear from the figure that the models do not provide an accurate prediction for the creep coefficient.

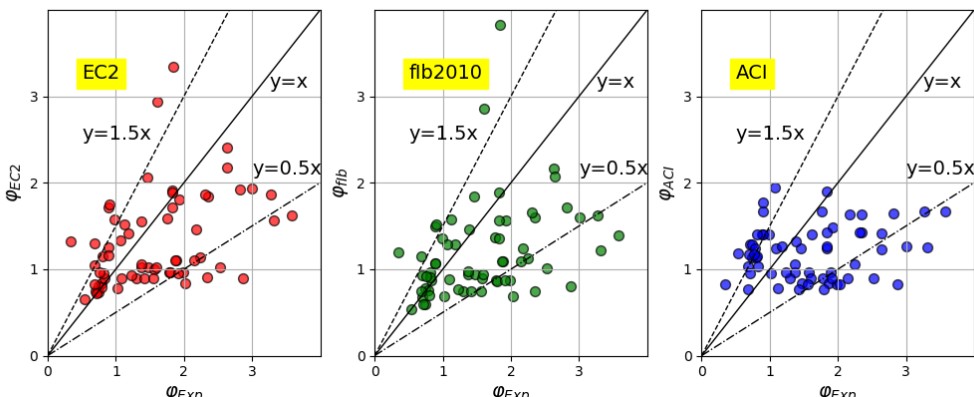

**Figure 1.** Experimental versus predicted creep coefficients for natural aggregates concrete (NAC).

To statistically quantify the performance of each model, the covariance, cov, and the correlation factor, $R^2$, were calculated according to the following equations:

$$Cov\left(\varphi_{exp}, \varphi_{cal}\right) = \frac{1}{N} \sum \left(\varphi_{exp} - \overline{\varphi}_{exp}\right)\left(\varphi_{cal} - \overline{\varphi}_{cal}\right)$$

(3)

$$R^2 = 1 - \frac{SSE}{SST} = 1 - \frac{\sum\left(\varphi_{exp} - \varphi_{cal}\right)^2}{\sum\left(\varphi_{exp} - \overline{\varphi}_{exp}\right)^2}$$

(4)

with

$SSE = \sum\left(\varphi_{exp} - \varphi_{cal}\right)^2$ the sum of squares of error,

$SST = \sum\left(\varphi_{exp} - \overline{\varphi_{exp}}\right)^2$ the deviations of the experimental points from their mean.

Table 2 shows the statistical results for all models and indicates EC2 as the most suitable model for the prediction of creep, followed by ACI 209.2R-08 and fib MC2010. Therefore, only the EC2 model was analyzed in detail, with the aim of studying the possible modification to improve the creep prediction of NAC and taking into account the presence of recycled aggregates in the modified expressions.

**Table 2.** Correlations between experimental and calculated creep coefficients.

|  | Mean φ | Covariance, Cov | $R^2$ |
| --- | --- | --- | --- |
| Experimental | $1.49 \pm 0.7$ | - | - |
| EC2 | $1.62 \pm 0.82$ | 0.093 | 0.49 |
| ACI 209.2R-08 | $1.21 \pm 0.30$ | 0.043 | 0.37 |
| MC 2010 | $1.60 \pm 1.01$ | 0.068 | 0.39 |

The extreme values above the line $y = 1.5x$ and below the line $y = 0.5x$ were sorted and the differences between the predicted and experimental values were calculated. The obtained results show that the deviations are independent of the variables considered in the current models, as shown in Appendix A Figure A2 for EC2.

Figure 2 shows experimental versus predicted creep of concrete mixtures by Fathifazl et al. [8], Geng et al. [40], and Kou et al. [34].

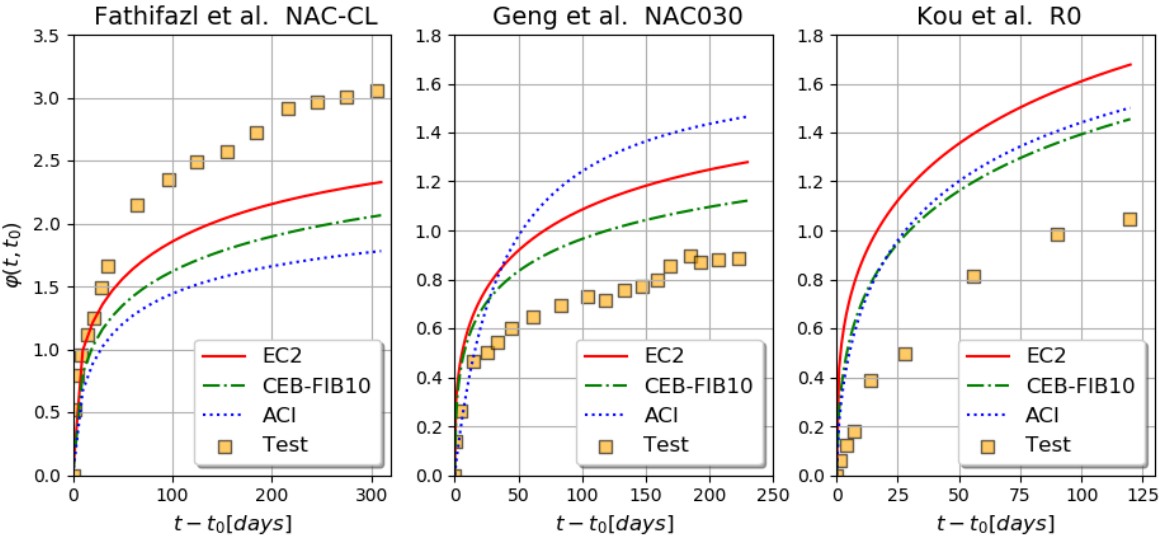

**Figure 2.** Experimental and predicted creep development with time according to the empirical equations presented in Table 1 for several studies [8,40]. It is obvious that the models in their current state are unable to predict the evolution of the creep coefficient. The analytical models underestimate the behavior of the concretes studied by Fathifazl et al. [8] and overestimate it in the studies of Kou et al. [34] and Geng et al. [40].

### 3.1. Creep Parameters' Sensitivity of EC2 Model

In order to understand the origin of the dispersion between the experimental and the predicted results, the sensitivity of the parameters describing the EC2 model was studied. The sensitivity study was started by verifying the effect of the compressive strength, $f_{cm}$, and the notional size, $h_0$, on the parameter $\varphi_{RH}$, which takes into account the effect of relative humidity on the notional creep coefficient $\varphi_0$. From the general equation shown in Table 1, the two parameters can be defined mathematically as:

$$\varphi_{RH} = \left[1 + \frac{1 - \frac{RH}{100}}{0.1\sqrt[3]{h_0}}\right] \text{ for } f_{cm} \leq 35 \text{MPa}$$

$$\varphi_{RH} = \left[\left(1 + \frac{1 - \frac{RH}{100}}{0.1\sqrt[3]{h_0}}\left[\frac{35}{f_{cm}}\right]^{0.7}\right)\left[\frac{35}{f_{cm}}\right]^{0.2}\right] \text{ when } f_{cm} > 35 \text{MPa}$$

(5)

and

$$\varphi_0 = \varphi_{RH}\frac{16.8}{\sqrt{f_{cm}}}\frac{1}{(0.1 + t_0^{0.20})}$$

(6)

The results illustrated in Figure 3 show that for a given class of compressive strength, this parameter decreases when the relative humidity increases, but it is not very sensitive with respect to the notional size ($h_0$).

Similarly, the sensitivity of $\beta_h$, which intervenes in the evaluation of the aging function, was evaluated. $\beta_h$ is given by Equation (7) and depends on the relative humidity as well as the notional size.

$$\beta_h = 1.5\left[1 + (0.012RH)^{18}\right]h_0 + 250 \leq 1500 \qquad f_{cm} \leq 35$$

$$\beta_h = 1.5\left[1 + (0.012RH)^{18}\right]h_0 + 250\alpha_3 \leq 1500\alpha_3 \qquad f_{cm} > 35$$

(7)

with $\alpha_3 = \left[\frac{35}{f_{cm}}\right]^{0.5}$.

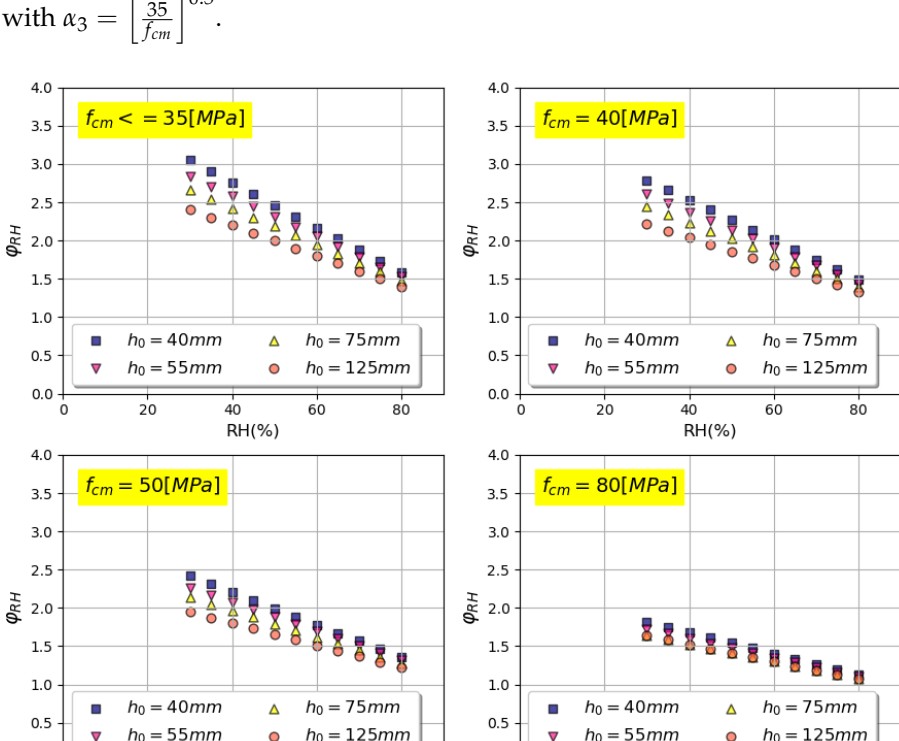

**Figure 3.** Variation of $\varphi_{RH}$ with relative humidity.

As can be seen in Figure 4, at a given compressive strength and a notional size, this parameter is very insensitive to the variation in relative humidity (*RH*), and it is also not at the origin of the observed dispersion between the experimental and the calculated values.

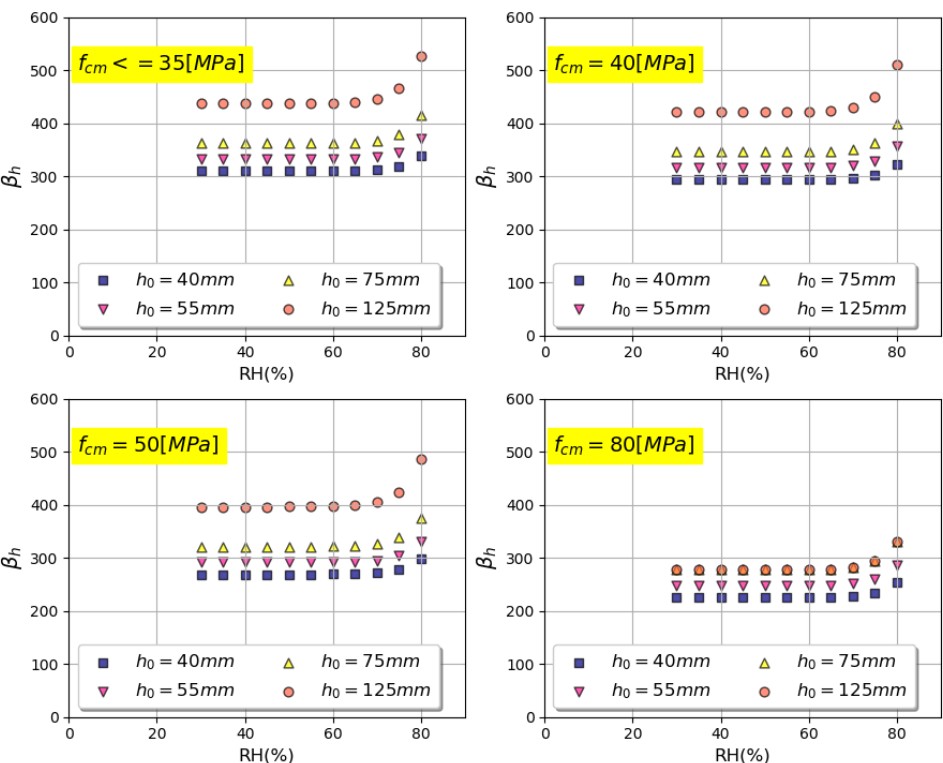

**Figure 4.** Variation of $\beta_h$ with relative humidity.

### 3.2. Identification of Creep Parameters

The creep model can be expressed by the following equation:

$$\varphi(t, t_0) = \varphi_{0,EXP} \left[ \frac{t - t_0}{\beta_h + t - t_0} \right]^{\alpha_{EXP}} \tag{8}$$

The equation has three parameters which can be at the origin of the dispersion between the experimental values and those predicted according to analytical models. Those terms are, coefficient of conventional creep, $\varphi_{0,EXP}$, the power of the aging function, $\alpha_{EXP}$, and $\beta_h$, which depends on the relative humidity ($RH\%$) and the notional member size, $h_0$. In this work, the three parameters were calculated using an optimization approach which consists in minimizing the difference between experimental creep curves and the calculated analytical ones according to Equation (8). The procedure was implemented in Matlab and begins by imposing initial values to parameters $\varphi_{0,EXP}$, $\alpha_{EXP}$, and $\beta_h$, allowing to generate a creep curve using Equation (8). The function fminsearch of Matlab is used iteratively to estimate new values, while the square of difference between analytical and experimental curves is higher that $10^{-3}$. The optimization was initially conducted for the NAC in order to verify the reliability of the analytical relationships, especially the EC2. The results illustrated in Figure 5 show that the average value of $\alpha_{EXP}$, the power of the aging function, is $0.44 \pm 0.15$ and not 0.3, as given in EC2.

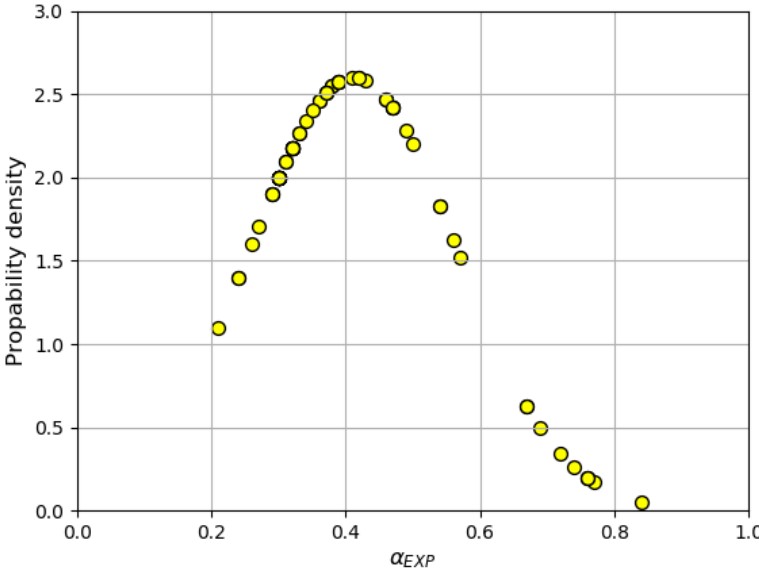

**Figure 5.** Distribution of $\alpha_{EXP}$, the power of the aging curve.

The sensitivity of $\varphi_{0,EXP}$ was studied with the variation of material parameters as well as the conditions of the creep test. The results are presented in Figure 6 as a function of: (a) concrete compressive strength, $f_{cm}$, (b) temperature, T, (c) relative humidity, $RH$, (d) notional member size, $h_0$, (e) stress level, $k_\sigma$, (f) the age of concrete at loading, $t_0$, (g) paste volume, $V_{paste}$, and (h) sand to aggregate ratio, $\frac{S}{S+G}$. Indeed, no correlation is clear except the correlation with sand to aggregate ratio, where $\varphi_{0,EXP}$ increases when $\frac{S}{S+G}$ increases (Figure 6h). Taking this observation into account, a new expression for the coefficient of conventional creep was proposed as follows:

$$\varphi_{0,\text{modif}} = 5.7 \left( \frac{S}{S+G} \right)^2 \varphi_{0,EC2} \tag{9}$$

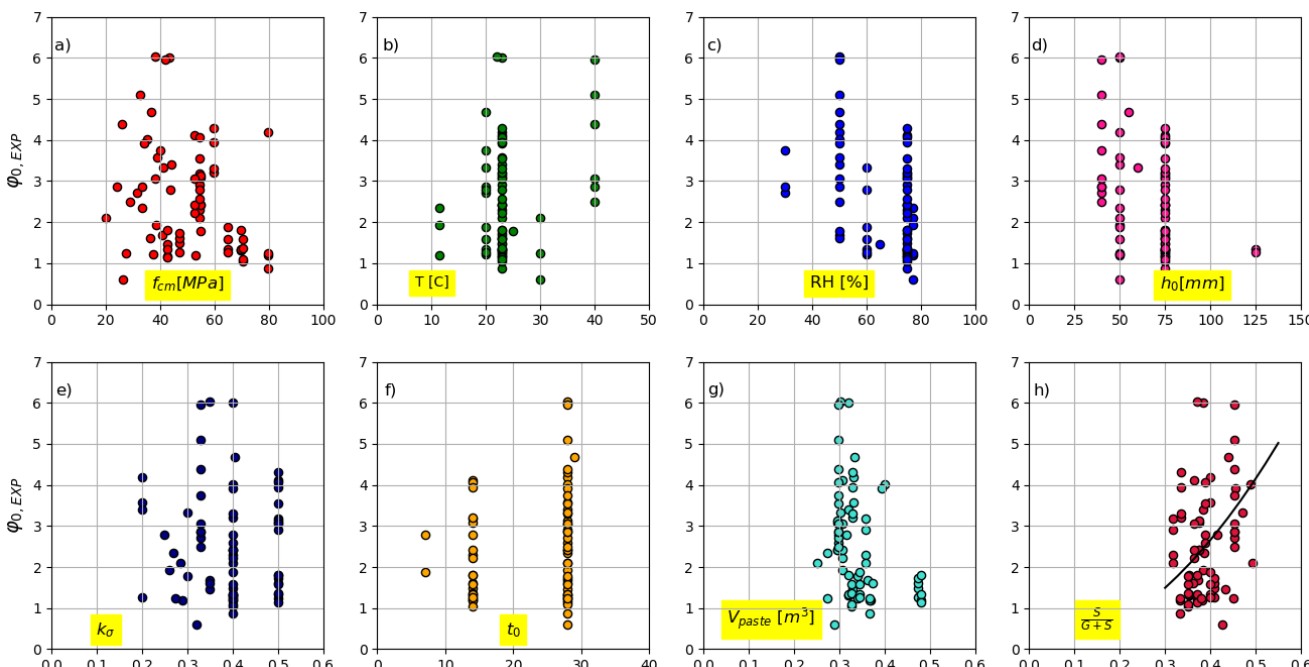

**Figure 6.** The sensitivity of $\varphi_{0,EXP}$ versus: (**a**) compressive strength, (**b**) temperature, (**c**) relative humidity, (**d**) notional size, (**e**) stress level, (**f**) loading age, (**g**) paste volume, sand to aggregate content, (**h**) sand by aggregates(G + S) ratio.

The optimized parameter $\beta_{h.EXP}$ was analyzed according to the EC2 parameters. Those include the compressive strength, $f_{cm}$, the relative humidity, RH, and the notional size, $h_0$. The results illustrated in Figure 7 show that for a given notional size and in a controlled environment, $\beta_{h.EXP}$ increases with the increase in $f_{cm}$. Based on this observation, the initial expression given by EC2 as expressed in Equation (7) was modified into Equation (10) by explicitly introducing the compressive strength, $f_{cm}$, based on 73 values of NAC.

$$\beta_{h,Modif} = 1.5\Big[1 + (0.012RH)^{18}\Big]h_0 + mf_{cm}^3 \tag{10}$$

with $m = 7.5 \times 10^{-4}$.

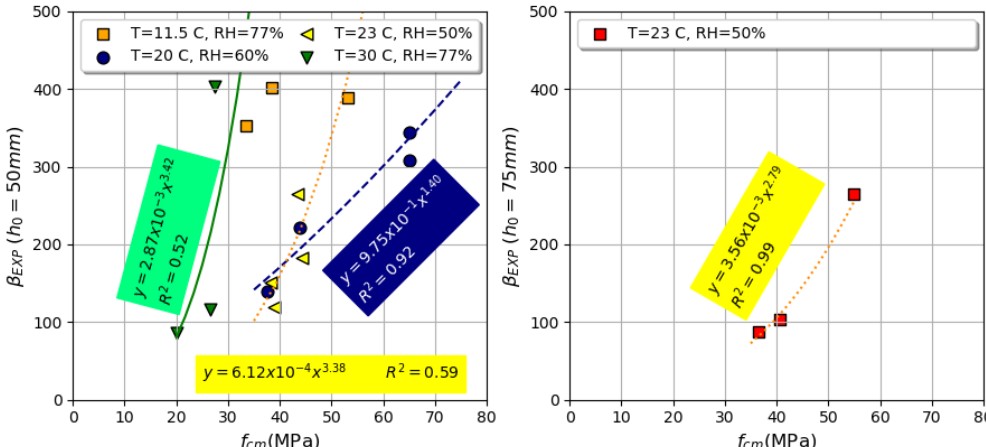

**Figure 7.** Evolution of $\beta_{h.EXP}$ with $f_{cm}$ for given relative humidity and notional size.

Figure 8 represents a comparison between the optimized values, $\beta_{h.EXP}$, with the values calculated according to Equation (7) (Figure 8a) and the values calculated with the modified Expression (10) (Figure 8b). Although the correlation is better when Equation (10) is used instead of Equation (7), the number of points does not seem sufficient yet to express the correlation coefficient.

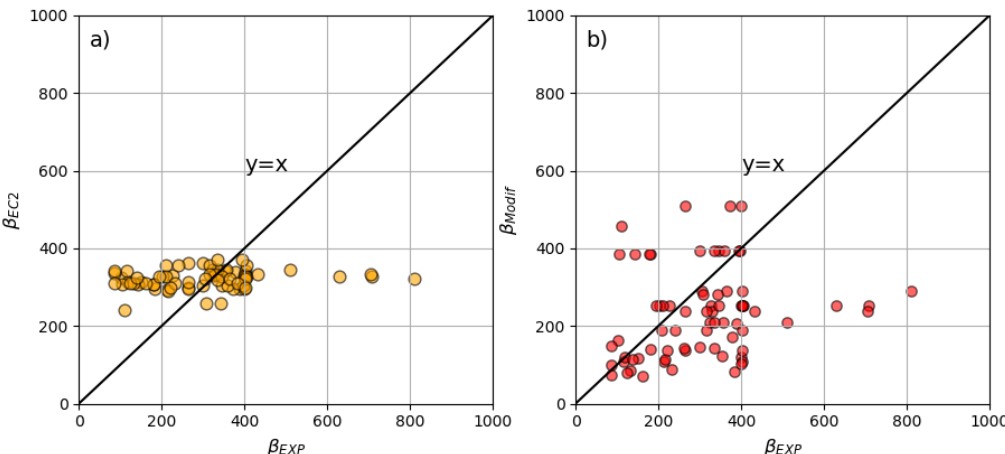

**Figure 8.** (**a**) Optimized $\beta_{h.EXP}$ versus $\beta_{h.EC2}$, (**b**) optimized $\beta_{h.EXP}$ versus modified $\beta_{h.Modif}$.

By considering all the proposed modifications, a new relation to predict the coefficient of creep can be written as:

$$\varphi(t,t_0) = 5.7 \left[ \frac{S}{S + G} \right]^2 \varphi_{0,EC2} \left[ \frac{t - t_0}{\beta_{h\_Modif} + t - t_0} \right]^{0.44} \tag{11}$$

Using the two modified parameters $\varphi_{0.Modif}$ and $\beta_{h.Modif}$, a modification of the creep model of ACI 209.2R–08 can be proposed as follows:

$$\varphi(t,t_0) = \varphi_{0,Modif} \frac{(t - t_0)^{\psi}}{\beta_{h,Modif} + (t - t_0)^{\psi}} \tag{12}$$

An optimization approach consisting in reducing the differences between experimental creep curves and the ones predicted by Equation (12) allowed to find an average value $\psi = 1.2$.

The creep coefficients for materials presented in Figure 2 were recalculated using Equations (11) and (12) and the results are illustrated in Figure 9. It can be observed that the calculated values agreed well with experimental data on the development of creep over time. The correlation coefficients between the experimental and the calculated values using the modified models for the modified EC2 (Equation (11)) and the modified ACI (Equation (12)) respectively, are given in Table 3. The detailed analysis is given in Appendix A Table A6.

**Table 3.** Correlations between experimental and calculated creep coefficients using modified models.

| Modified Models | Mean $\varphi$ | Covariance, Cov | $R^2$ |
|---|---|---|---|
| Modified EC2 | $1.21 \pm 0.69$ | 0.41 | 0.81 |
| Modified ACI 209.2R-08 | $1.06 \pm 0.77$ | 0.52 | 0.77 |

The model MC2010 was not modified due to lack of data related to the effect of recycled aggregates on basic and drying creep separately.

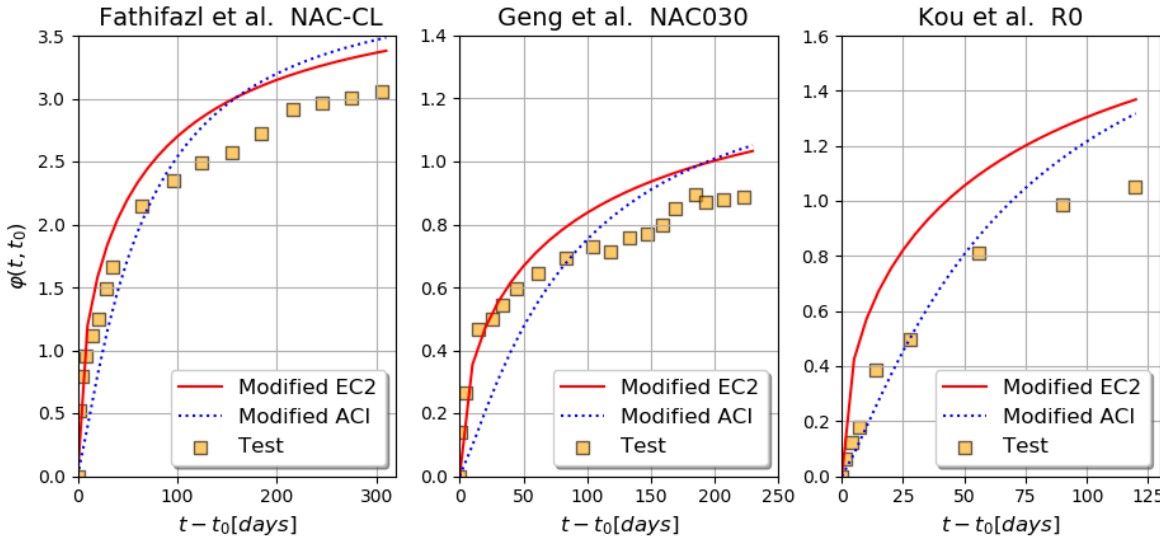

**Figure 9.** Experimental versus predicted creep using modified models of concrete mixtures by Fathifazl et al. [8], Geng et al. [40], and Kou et al. [34].

## 4. Creep Prediction of RAC

The validity of the standards presented in Table 1 was checked for RAC too. The results represented in Figure 10 and the correlation coefficients, $R^2$, summarized in Table 4 show that the prediction is not sufficiently acceptable to determine the creep coefficient of RAC. Nevertheless, the correlation coefficient is better for EC2 compared to ACI.

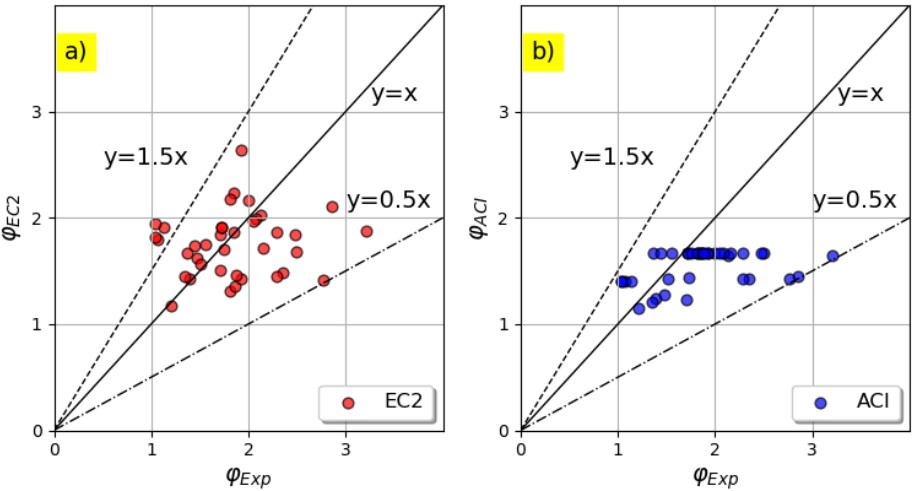

**Figure 10.** Experimental versus predicted creep coefficients for Recycled Aggregates Concrete (RAC) using: (**a**) EC2 model, (**b**) ACI 209.2R-08.

**Table 4.** Correlation factors for modified models.

|  | Without Modification | | | Modified Models | | |
|---|---|---|---|---|---|---|
|  | Mean | Cov | $R^2$ | Mean | Cov | $R^2$ |
| Experimental values | $1.43 \pm 0.93$ | | | | | |
| EC2 | $1.34 \pm 0.65$ | 0.49 | 0.62 | $1.36 \pm 0.81$ | 0.67 | 0.79 |
| ACI 209.2R-08 | $1.05 \pm 0.63$ | 0.51 | 0.56 | $1.13 \pm 0.93$ | 0.77 | 0.69 |
| CEB FIP 2010 | $1.2 \pm 0.63$ | 0.44 | 0.52 | - | - | - |

In order to account for the presence of recycled aggregates, the term $(1 + \alpha_M \Gamma_m)$ was introduced in Equation (11) and a value of $\alpha_M = 0.33$ was found by a linear regression. The final expression for creep coefficient for recycled aggregate concrete takes the form:

$$\varphi(t, t_0) = 5.7 \left[ \frac{S}{S + G} \right]^2 (1 + 0.33\Gamma_m) \varphi_{0,EC2} \left[ \frac{t - t_0}{\beta_{h\_Modif} + t - t_0} \right]^{0.44} \tag{13}$$

The corrective term $(1 + 0.33\Gamma_m)$ was also introduced in the expression of Equation (12) for the calculation of RAC creep coefficient according to the ACI standard. It is worth mentioning that for the qualification of the models for the RAC, certain studies were eliminated from the database. The works of Tia et al. [36] and Ghodousi et al. [37] have been omitted because the authors used a percentage of cementitious materials in concrete mixtures higher than 30% of the total mass of binder, which is the allowed limit defined by European standards [25,39]. For other studies [11,16,35], the creep values were too far from other values. Hence, a total of 20 RAC values were used for the validation of proposed modified relations. The predicted curves using modified models are compared to the creep data selected in the database of the present work. The correlation factors are tabulated in Table 4, where it is rather interesting to note that they are better than values obtained for models without modification.

Some representative comparisons for RAC before and after modification are presented in Figure 11. The difference without proposed modifications is explained by the fact that the normative models do not consider the effect of recycled aggregates on creep in addition to their insufficiency already mentioned for the NAC. By considering all modifications, the calculated values agreed well with experimental data on the development of creep over time. This improvement is attributed to the modification of $\beta_h$ and the power of the ageing function, as well as to the introduction of the term $\frac{S}{S + G}$ for NAC together with the introduction of $1 + 0.33\Gamma_m$ for RAC.

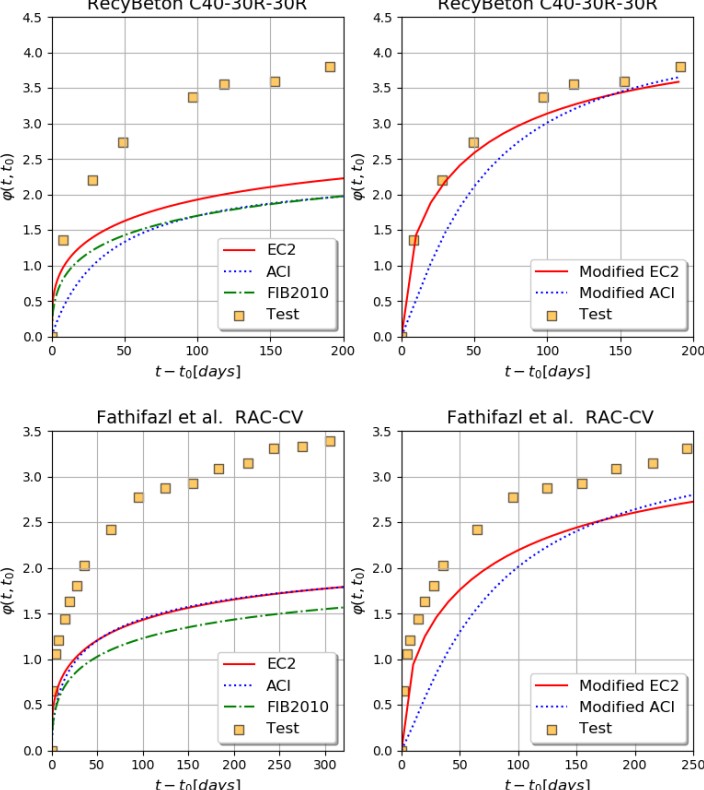

**Figure 11.** Experimental versus predicted creep using modified models for RAC mixtures by Fathifazl [8] and RecyBeton [30].

## 5. Conclusions

This study presented an analysis of data available in the literature on the ability of design standards to predict the creep behavior of natural and recycled aggregate concretes. The comparison between experimental and predicted values shows a significant dispersion in terms of creep coefficient. However, EC2 was the most suitable for estimating creep for NAC and RAC as compared to both ACI and MC2010 models and it was then chosen for a possible modification.

Based on the general expression of the creep function, the conventional creep coefficient, named $\varphi_{0,EXP}$, the power of the aging function, $\alpha_{EXP}$, as well as $\beta_h$ were identified by an optimization approach using the database of NAC. It was found that the value of $\alpha_{EXP}$, the power of the aging function, is $0.44 \pm 0.15$ and not 0.3, as given in EC2, while for the coefficient of conventional creep, $\varphi_{0,EXP}$, and $\beta_h$, new expressions have been proposed. When these two parameters were set, the power of the ACI ageing function was identified, and a value of 1.2 was found. With all these modifications, the correlation between the experimental and predicted values was enhanced for both EC2 and ACI models.

In order to account for the presence of recycled aggregates in the modified expression, the term $(1 + 0.33\Gamma_m)$ has been considered with $\Gamma_m$, the equivalent replacement ratio, which represents the mass of recycled aggregates in the granular skeleton.

**Author Contributions:** Conceptualization, formal analysis, investigation, P.P.T.; supervision, validation and draft preparation, E.G. and G.W. All authors have read and agreed to the published version of the manuscript.

**Funding:** This research was financially supported by the French national project ANR ECOREB: (ANR-12-VBDU-0003; November 2012–November 2016, http://ecoreb.fr, accessed on 1 November 2012) which is highly appreciated.

**Data Availability Statement:** The data presented in this study are available on request from the corresponding author.

**Conflicts of Interest:** The authors declare no conflict of interest.

## Abbreviations

| | |
|---|---|
| $\varepsilon_\sigma$ | Elastic strain |
| $\varepsilon_c$ | Creep strain |
| $\varepsilon_{c\sigma}$ | Elastic and creep strain |
| $\varphi(t,t_0)$ | Creep coefficient |
| $J(t,t_0)$ | Compliance |
| $\sigma_c(t_0)$ | Constant compressive stress |
| $h_0$ | Notional size of the member (mm) |
| $f_{cm}$ | Mean compressive strength of concrete at the age of 28 days (MPa) |
| $t_0$ | Age of concrete at loading (days) |
| $t_{0,adj}$ | Adjusted age of loading (days) |
| $t$ | Age of concrete at the moment considered (days) |
| NAC | Natural aggregate concrete |
| Wa(%) | Water absorption of the aggregates. |
| $E_{cm}$ | Secant elastic modulus |
| $E_{cm}(t_0)$ | Secant elastic modulus at the moment of creep loading |
| $\gamma_{c,t0}$ | Age of loading correction factor |
| $\gamma_{c,RH}$ | Relative humidity correction factor |
| $\gamma_{c,VS}$ | Volume-surface ratio correction factor |
| $\gamma_{c,S}$ | Slump correction factor |
| $\gamma_{c,\Psi}$ | Fine aggregate correction factor |
| $\gamma_{c,\alpha}$ | Air content correction factor |
| V/S | Volume-surface ratio |
| $\beta_H$ | Coefficient depending on the relative humidity |
| RH | Relative humidity (%) |
| W/B | Water to binder ratio |

| | |
|---|---|
| $\Gamma_m$ | Mass equivalent substitution ratio. |
| RAC | Recycled aggregate concrete |
| RCA | Recycled concrete aggregates |
| $h_0$ | Notional size |

## Appendix A

**Table A1.** Required parameters for creep models.

| N° | Parameter | ACI 209R-08 | EC2 | MC 2010 |
|----|-----------|-------------|-----|---------|
| 1 | Concrete strength at 28 days | | x * | x |
| 2 | Type of cement | x | x | x |
| 3 | Curing type | x | | |
| 4 | Relative humidity | x | x | x |
| 5 | Curing temperature | | x | x |
| 6 | Age of concrete at loading | x | x | x |
| 7 | Stress level | | x | x |
| 8 | Age of concrete at curing | | | |
| 9 | Volume-surface ratio | x | | |
| 10 | Slump | x | | |
| 11 | Percentage of fine aggregates by weight | x | | |
| 12 | Percentage of air content by volume | x | | |
| 13 | Cross sectional area | | x | x |
| 14 | perimeter of the section in contact with the atmosphere | | x | x |

* The letter x means required parameter.

**Table A2.** Fields of validity of creep models.

| N° | Variable | ACI 209.2R-08 | EC2 MC 2010 |
|----|----------|---------------|-------------|
| 1 | Type of cement | Type I and III | - |
| 2 | Slump | 70 mm | - |
| 3 | Air content | ≤6% | - |
| 4 | % Fine aggregate | 50% | - |
| 5 | Cement content | 279 to 446 kg/m$^3$ | - |
| 6 | Length of initial curing (Moist cured) | 7 days | - |
| 7 | Length of initial curing (Steam cured) | 1 to 3 days | - |
| 8 | Curing temperature | 23.2 ± 2 °C | - |
| 9 | Relative humidity (Curing) | ≥95% | - |
| 10 | Concrete temperature | 23.2 ± 2 °C | - |
| 11 | Relative humidity | 40% | - |
| 12 | Volume/Surface | V/S = 38 mm | - |
| 13 | Minimum thickness | 150 mm | - |
| 14 | Concrete age at load application (Moist-cured) | 7 days | - |
| 15 | Concrete age at load application (Steam-cured) | 1 to 3 days | - |

**Table A2.** *Cont.*

| N° | Variable | ACI 209.2R-08 | EC2 MC 2010 |
|----|----------|---------------|-------------|
| 16 | During of loading period | Sustained load | Sustained load |
| 17 | Compressive stress | Axial compression | Axial compression |
| 18 | Stress/strength ratio | $\leq 0.50$ | $\leq 0.50$ |

**Table A3.** Selected models for creep prediction of recycled aggregate concrete.

| Author | Year | Studied Models | Suggested Modification |
|--------|------|----------------|------------------------|
| Fathifazl et al. [8]' | 2011 | Neville CEB-FIB90 EC2 | Introduction of residual mortar volume fraction |
| Fan et al. [19] | 2014 | Neville | Introduction of residual mortar volume fraction |
| Fathifazl and Razaqpur [18] | 2013 | ACI 209.2R–08 4 rheological models | RCA coefficient taking into account the presence of attached mortar |
| Silva et al. [20] | 2015 | EC2. ACI 209-R. Bazant B3 and GL2000 | Correction factor for creep coefficient |
| Knaack and Kurama [15] | 2015 | ACI 209R | Modification of the ultimate creep by a correction factor |
| Lye et al. [17] | 2015 | ACI 209. CEB-FIB90. EC2. GL2000. Bazant B3and Hong Kong Code | Correction factor based on RCA content |
| Geng et al. [21] | 2016 | Creep coefficient (Equation (3)) | Correction using density or water absorption of RCA |
| Pan and Meng [22] | 2016 | CEB-FIP 90 | Modification of the ultimate creep coefficient and the power of time development function |
| Tošic et al. [9] | 2019 | MC 2010 | Correction factor based on RCA ratio |
| Geng et al. [23] | 2019 | Neville | Correction factors based on the residual cement paste |

**Table A4.** Database for creep analysis.

| Author | Year | Type of Cement | W/B | S/(S + G) | $t_0$ (Days) | Load Duration (Days) | Load Level | RH (%) | Γm |
|--------|------|----------------|-----|-----------|------|----------------------|------------|--------|-----|
| Sriravindrarajah and Ravindrarajah and Tam [35] | 1985 | Portland Type I (28) | 0.51–0.73 | 0.43–0.49 | 28 | 56 | 0.27–0.32 | 77 | 0–0.51 |
| Tia et al. [36] | 2005 | Portland Type I (28) | 0.24–0.44 | 0.32–0.41 | 14, 28 | 91 | 0.4–0.5 | 75 | - |
| Kou et al. [34] | 2007 | Portland Type I (28) | 0.55 | 0.36–0.37 | 28 | 120 | 0.35 | 50 | 0–0.61 |
| Domingo-Cabo et al. [11] | 2009 | CEM I 42.5 N/SR | 0.50 | 0.433 | 28 | 180 | 0.35 | 65 | 0–0.55 |
| Ghodousi et al. [37] | 2009 | Portland Type II (28) | 0.43 | 0.454 | 28 | 120 and 200 | 0.33 | 30, 50 | - |
| Fathifazl et al. [8] | 2011 | Portland Type I (28) | 0.45 | 0.46–0.49 | 28 | 330 | 0.4 | 50 | 0–0.50 |
| Manzi et al. [32] | 2013 | CEM II-A/LL 42.5 R | 0.48 | 0.471 | 28 | 500 | 0.3 | 60 | - |
| Tang et al. [33] | 2014 | Portland Type I (28) | 0.50 | 0.35 | 28 | 270 | 0.3 | 50 | - |
| Knaack and Kurama [15] | 2015 | Portland Type I (28) | 0.44 | 0.383 | 28 | 240 | 0.2–0.4 | 50 | 0–0.59 |
| Surya et al. [38] | 2015 | Portland Type I (28) | 0.40 | 0.371 | 28 | 90 | 0.35 | 50 | 0–0.61 |
| Zhao et al. [29] | 2016 | Portland (28) | 0.45 | 0.38 | 28 | 360 | 0.4 | 60 | - |
| Recybeton [30] | 2016 | CEM II/A-L 42.5 N | 0.49 | 0.44 | 28 | 200 | 0.41 | 50 | 0–0.52 |
| Geng et al. [21] | 2016 | Portland Type I (28) | 0.3–0.6 | 0.38–0.39 | 28 | 230 | 0.26–0.29 | 77 | 0–0.64 |
| Pan and Meng [22] | 2016 | - | 0.33 | 0.4 | 7, 28 | 500 and 600 | 0.2–0.4 | 60 | - |
| Gholampour and Ozbakkaloglu [16] | 2018 | Portland Type I (28) | 0.27–0.50 | 0.4 | 28 | 450 | 0.2 | 50 | 0–0.57 |
| He et al. [31] | 2017 | - | 0.40 | 0.415 | 7 | 180 | 0.25 | 60 | - |

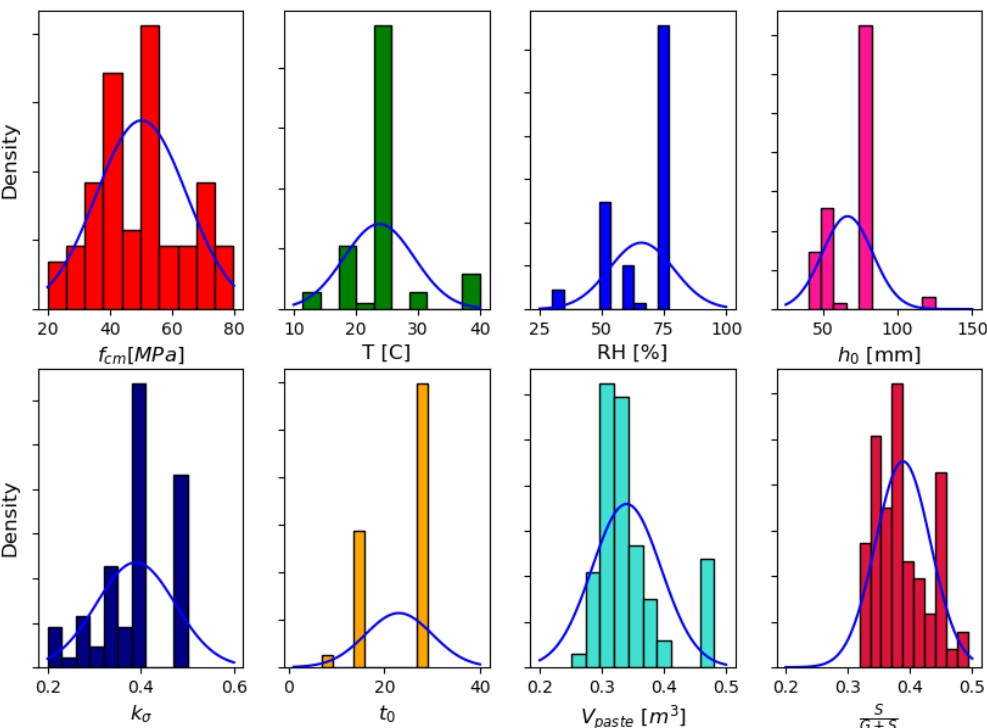

**Figure A1.** Histograms of statistical distribution of variables.

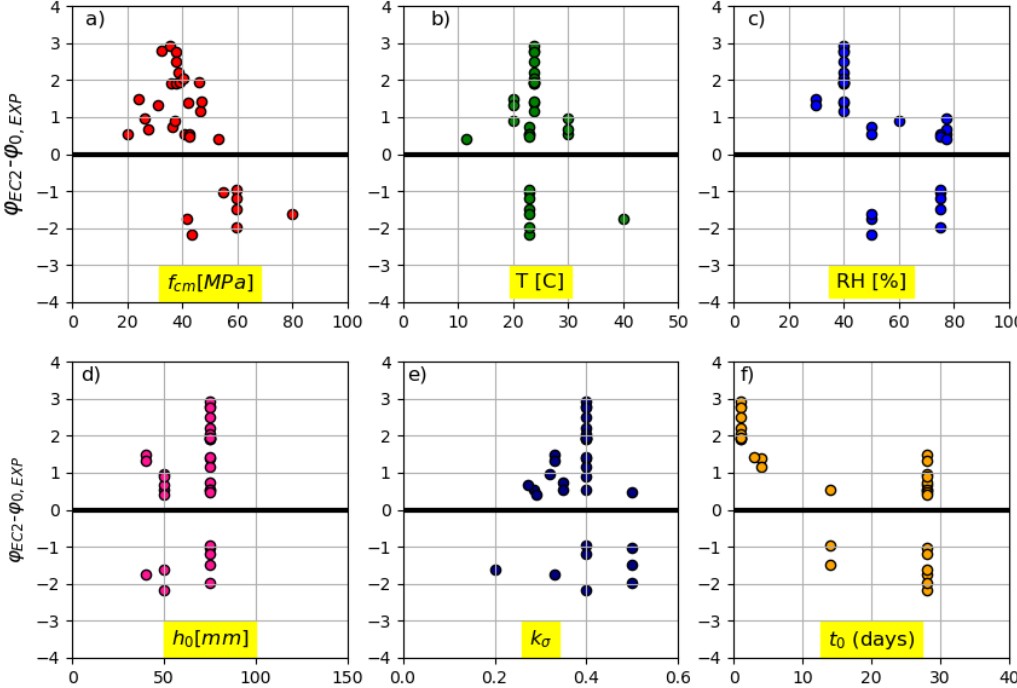

**Figure A2.** Difference between experimental and EC2 predicted values as a function of: (**a**) compressive strength, (**b**) temperature, (**c**) relative humidity, (**d**) notional size, (**e**) stress level, (**f**) loading age.

**Table A5.** Ranges of studied parameters.

| Input Variables | NAC Values (73) | RAC Values (48) |
|---|---|---|
| $f_{cm28}$ (MPa) | 20.0–79.8 | 18.5–80.1 |
| w/c | 0.24–0.73 | 0.29–0.75 |
| Wa. (%) | 0.44–4.18 | 1.89–4.55 |
| Cement content (kg/m$^3$) | 117–600 | 275–600 |
| Type of cement | Portland Type I/Portland Type II CEM II/A-L 42.5 N/ CEM I 42.5 N/SR | Portland Type I/ CEM II/A-L 42.5 N/ CEM I 42.5 N/SR |
| Curing time | >7 days | >7 days |
| $t$–$t_0$ (days) | 56–600 days | 56–450 days |
| $t_0$ (days) | 7–28 days | 7–113 days |
| Loading level | 0.2–0.5 | 0.2–0.43 |
| $h_0$ (mm) | 40–125 | 50–75 |
| $RH$ (%) | 30–77 | 50–77 |
| T (°C) | 11.5–40 | 11.5–30 |

**Table A6.** Detailed creep analysis of NAC-CL [8].

| Time (Day) | Experimental | EC2 | MC2010 | ACI | Modified EC2 Equation (11) | Modified ACI Equation (12) |
|---|---|---|---|---|---|---|
| 0 | 0.00 | 0.00 | 0.00 | 0.00 | 0.00 | 0.00 |
| 10 | 1.01 | 0.99 | 0.81 | 0.67 | 1.20 | 0.39 |
| 20 | 1.23 | 1.21 | 1.02 | 0.88 | 1.58 | 0.80 |
| 30 | 1.54 | 1.36 | 1.16 | 1.02 | 1.84 | 1.16 |
| 40 | 1.76 | 1.47 | 1.26 | 1.12 | 2.04 | 1.46 |
| 50 | 1.95 | 1.56 | 1.35 | 1.20 | 2.20 | 1.72 |
| 60 | 2.09 | 1.64 | 1.42 | 1.27 | 2.33 | 1.94 |
| 70 | 2.20 | 1.70 | 1.48 | 1.32 | 2.44 | 2.12 |
| 80 | 2.27 | 1.76 | 1.53 | 1.37 | 2.54 | 2.28 |
| 90 | 2.32 | 1.81 | 1.58 | 1.41 | 2.62 | 2.42 |
| 100 | 2.37 | 1.86 | 1.62 | 1.44 | 2.70 | 2.54 |
| 110 | 2.42 | 1.90 | 1.66 | 1.47 | 2.76 | 2.64 |
| 120 | 2.47 | 1.93 | 1.69 | 1.50 | 2.82 | 2.73 |
| 130 | 2.51 | 1.97 | 1.72 | 1.53 | 2.88 | 2.81 |
| 140 | 2.53 | 2.00 | 1.75 | 1.55 | 2.93 | 2.89 |
| 150 | 2.55 | 2.03 | 1.78 | 1.57 | 2.97 | 2.95 |
| 160 | 2.59 | 2.06 | 1.81 | 1.59 | 3.01 | 3.01 |
| 170 | 2.63 | 2.09 | 1.83 | 1.61 | 3.05 | 3.06 |
| 180 | 2.69 | 2.11 | 1.85 | 1.63 | 3.09 | 3.11 |
| 190 | 2.76 | 2.13 | 1.88 | 1.64 | 3.12 | 3.16 |
| 200 | 2.82 | 2.15 | 1.90 | 1.66 | 3.15 | 3.20 |
| 210 | 2.88 | 2.17 | 1.91 | 1.67 | 3.18 | 3.23 |

**Table A6.** *Cont.*

| Time (Day) | Experimental | EC2 | MC2010 | ACI | Modified EC2 Equation (11) | Modified ACI Equation (12) |
|---|---|---|---|---|---|---|
| 220 | 2.92 | 2.19 | 1.93 | 1.69 | 3.20 | 3.27 |
| 230 | 2.94 | 2.21 | 1.95 | 1.70 | 3.23 | 3.30 |
| 240 | 2.95 | 2.23 | 1.97 | 1.71 | 3.25 | 3.33 |
| 250 | 2.97 | 2.24 | 1.98 | 1.72 | 3.27 | 3.36 |
| 260 | 2.98 | 2.26 | 2.00 | 1.73 | 3.29 | 3.38 |
| 270 | 3.00 | 2.27 | 2.01 | 1.74 | 3.31 | 3.40 |
| 280 | 3.02 | 2.29 | 2.03 | 1.75 | 3.33 | 3.43 |
| 290 | 3.04 | 2.30 | 2.04 | 1.76 | 3.35 | 3.45 |
| 300 | 3.05 | 2.32 | 2.05 | 1.77 | 3.36 | 3.47 |

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
