# Peer review of "Towards a New Analytical Creep Model for Cement-Based Concrete Using Design Standards Approach"

_buildings, doi:10.3390/buildings11040155_

Round 1

Reviewer 1 Report

Dear Authors,

I have reviewed your paper and I found it really interesting. I noted that this document does contain a solid review of the related literature on the topic and that you have something to contribute to that area of knowledge. 

The first aim of the paper to revise the applicability of analytical models adopted in design standard for NAC and to study their possible extension to RAC using a new database built from the experimental results available in the literatura turns out, a priori, very interesting.

There are some minor errors that should be correct, as for example:

  1. many databases have been proposed in the bibliographical references, but unfortu- 154 -> Many should be capital letter.
  2. mod-ification -> line192
  3. accord- ing -> line 205
  4. The graphs of Figure 6 have been exported in poor quality, they must be changed to look properly.
  5. There are 2 Figure 7. They must correct this error and that of the other figures consecutively.
  6. Recurrently throughout the text the equations that are inserted in the text have a wrong alignment.

Here are some things that need to be corrected, in my opinion, for publication:

  • Choosing a unique color criterion for all charts: Figure 2 would have been easier to understand if the color criterion set for Figure 1 (red, blue, and green) had been followed. I suggest you choose one ACI ara color, one for EC" and another for fib and be consistent with it to the end. This will make it easier to compress the graphs, which is as interesting a topic as it is complex.
  • In the graphs, and therefore in their conclusions, the phenomenon so disparate of both references must be further explained. Sólo se dice: "The analytical models underestimate the behavior of the concretes studied by 208 Fathifazl et al. [8] and overestimate it in the studies of Geng et al. [41]. "
  • Figure 7 (the first one): What are the adjustment parameters for the curves in Figure 7?. They do not seem too well adjusted (it is necessary to raise the graph differently because the points under the numerical expressions of the settings do not look good). In addition, one of the datasets to be adjusted is missing, and some of them, such as yellow or green, do not appear to have adequate regression. Is it possible to indicate the R parameter of the setting?
  • Figure 10. Experimental versus predicted creep using modified models for RAC mixtures by [8] and RecyBeton [31] -> Why the difference between experimental results (in yellow tables) and the rest? This should be explained in the text in greater depth.

The conclusions are fine, but they can be improved and explained in a more orderly manner.

I encourage you to correct these errors, and to give a paused reading correcting other possible errors in word splitting, as the criterion chosen by your word processor does not seem correct. In general, the graphs need a revision, some of the aspects I have explicitly discussed in this review.  I'll be happy to re-check this job.

Author Response

The authors would like to thank the reviewer for his constructive remarks which led to improve the quality of the manuscript. The detailed answers are given in the attached file.

The corrections are made in blue in the revised text

Reviewer 2 Report

Paper “Towards a new analytical creep model for cement based concrete using design standards approach”: The addressed topic is focal and of great interest for Buildings readers. It is in line with some of its subject areas such as building materials.

The aim of this work is to study the validity of the analytical models proposed in the most used international standards to evaluate the  creep properties and to check the possibility of their extension to estimate the creep of recycled aggregates concrete (RAC). 

It can contribute to a better understanding of the creep behavior of cement and  can be very useful for researchers in the same field.  

Manuscript style: It is necessary to review formally the whole manuscript, paying particular attention to punctuation, because sometimes it is very difficult to understand what you mean

Contents:

Line 90: check the subscripts of the terms used in the equations: t0

Line 103: Write the equations within the text of the paragraph.

Line 153: use capital letters to write the first word of the paragraph

Line 165:  what is the time of the creep curves studied in the experimental database section? it should be 90 days to be consistent with what you say on line 180, …creep time curves after 90 days

Lines 242-243 it is difficult to understand revise it please.

Lines 264-264: Revise the punctuation, it is difficult to understand

Line 369: Chek the sentence,  as it is currently written does not make sense.

Lines 224-225: try to fit the formulas into the paragraph. In this way they complicate the reading of it

Line 368: Graphics should be modified to make them look better

Line 371: Check the position of the subscripts in the variables

Line 402: why do not use you the model CEB FIP 2010? Explain it

Lines 425 ,429, 448: try to fit the formulas into the paragraph. In this way they complicate the reading of it

Lines 483-484: Revise the punctuation, it is difficult to understand

Author Response

The authors would like to thank the reviewer for his constructive remarks which led to improve the quality of the manuscript. The detailed answers are given in the attached file.

The corrections are made in red in the revised text

Reviewer 3 Report

The paper aims to improve analytical expressions to predict creep coefficient in normal aggregate concrete as well as on recycled aggregate concrete. In the study expressions provided by EC2, ACI, by Fib Model Code 90 and  by Fib Model Code 2010, arriving to some modified expression of relevant parameters.

In the reviewer opinion the paper needs to be improved according to the following remarks:

  1. What is the original contribution of the Authors?
  2. Authors declare that relevant parameters have been optimized, but the optimization algorithm is not described, For example, it is not clear what kind of distance (or distances) has been minimized in §3.2 and how many curves has been taken into account;
  3. Influence of water/concrete ratio and type of cement, if any, should be considered;
  4. Illustration of results is often lacking. Content of Figure 6, for example, is only partly described; but it clearly shows the dispersal of experimental results: some comments could be very helpful.
  5. Outcomes should be better discussed
  6. Introduction and conclusion should be amended on the base of the answers to previous remarks.
  7. The text needs some additional check and editing: in fact, in same cases punctuation is wrong and formulae in the text are frequently misaligned.
  8. Finally, since the differences between different predictions are not evident, it could be helpful to include in the annex an additional table where experimental creep coefficients are compared with those predicted by existing code or by the improved formulae. In that way it would possible to check the capability of each proposed approach to correctly identify the erxperimental results.

Author Response

The authors would like to thank the reviewer for his constructive remarks which led to improve the quality of the manuscript. The detailed answers are given in the attached file.

The corrections are made in pink in the revised text

Reviewer 4 Report

Dear authors,

The manuscript addresses two important issues, i.e. creep design in concrete and recycled aggregates concrete. According to the text, over 200 studies have been used to determine a new analytical creep model approach for both types of concrete. The notorious dispersion between experimental data and design standards have been clearly set.

But, with all due respect, the following comments should be taken into consideration.

1 The manuscript is full of typos (lines 40 and 41 are empty, capital letter missing in line 154, Table A.1, etc, “à” in Table A.2, the use of “.” instead of “,” in many lines, such as 231, 232, 265, etc.). Also “Figure 7” has been used twice, hence sequential numbering of figures in wrong. All unimportant things.

2 According to section 3, the initial approach is based on curves after 90 days of loading. I can assume that this is the best of what the scientific bibliography can offer, but seems insufficient in order to stablished a reformulation of standards. In my opinion, longer-term data are compulsory for this purpose. Moreover, the study of the different parameters involved in the EC design creep formulation is finally based on only two studies, [8] and [14], which, by all means, sounds insufficient in order to propose a modification in the standard. Furthermore, only 60 MPa have been plotted for concretes over 35 MPa.

3 Paragraph in line 242 assumes insensitivity but at low RH this does not seem correct for both fcm and ho.

4 Figure 6 shows up to 8 different plots, from a) to g), that should be described in the text. I am sorry but I cannot see the correlation in Figure 6.h. Please explain.

5 Please explain how did you modified equation 6 into equation 8 (obviously there is a typo there), but even assuming you are talking about equation 9, its modification only from Figure 6.h should be explained.

6 Please explain if formulas in Figure 7 (evolution) are determined by only the few points shown in the figure. Three or four points does not seem sufficient to the purpose.

7 Paragraph in line 381 indicates that “correlation is better” and that “the number of points does not seem sufficient”, which corroborates the aforementioned, as the figure shows dozens of points. The number of points is insufficient but the correlation, although could be considered as “better” is not good at all.

8 For all the aforementioned, the formulas proposed should not be considered as accurate, even though the improved correlation in Figure 8 (Experimental), as, again, is only based on two studies, hence, generalizations should be avoided.

9 The improvement indicated in lines 437-439 is negligible, specially the one shown for EC, which is the proposed for modification. The general modification of a standard should not be based on such small differences.

10 The improvement shown for RAC is once more based on only two studies.

Yours faithfully

Author Response

The authors would like to thank the reviewer for his constructive remarks which led to improve the quality of the manuscript. The detailed answers are given in the attached file.

The corrections are made in green in the revised text

Reviewer 5 Report

This paper presents in a first part a critical analysis of the main standards to evaluate the creep of conventional concretes. In the second part, it applies the conclusions obtained from the first part and from the bibliography to propose a new behavioural model for creep in recycled concretes.

In my opinion, this is a good work that uses a large number of experimental results and therefore makes the results valid for other situations. However, before it can be accepted, some modifications have to be made.    1.When trying to identify the reason for the dispersion of the results, I recommend that you apply the following technique:

     Select those data that are furthest away from the values proposed by the       standard.
     Then draw the histograms of best-fit, worst-fit above, worst-fit below for       all the variables in the dataset.
 In this way you will be able to identify if any of the variables in the dataset seem to have an influence on the outcome of the fit.

2. Lines 445 and 446, Remove some data for improper mix porportions (high percentage of cementitious materials). It is normal that if a mix porportion varies a lot from the others its resuts are out of the normal, but the paper should indicate the criteria to consider the mix proportions acceptable.

  I would also like to know whether the authors intend to publish the dataset they have created based on the results of other authors, perhaps as data in brief.

Author Response

The authors would like to thank the reviewer 5 for his  remarks which led to improve the quality of the manuscript. The detailed answers are given below

Reviewer 6 Report

Title:  Towards a new analytical creep model for cement based concrete using design standards approach

Manuscript Number: buildings-1118402-peer-review-v2

Reviewer's Comments
This Paper presents a literature review on the creep behavior using codes such as ACI209, EC2, CEB-FIP model code in order to develop a relationship allowing to predict the creep coefficient of natural and recycled aggregate concrete. This relationship is validated by experimental results collected from the literature. The paper is interesting for readers; however, minor corrections should be carried out before publication such as:

1- Page 3 line99: Define the symbol Ecm(to) in the text.

2- Subscripts of certain symbols should revised (i.e, page 5 line181)

3- Page 7, line 273-274: the sentence in these lines should be revised. The same remark in page 13, line 496.

4- Page 11, line 438: Notation should be corrected : αm or αM?

How the corrective term (1+0.33Γm) was obtained? Define αm and Γm in the text.

Author Response

The authors would like to thank the reviewer 6 for his  remarks which led to improve the quality of the manuscript. The detailed answers are given below

Reviewer 7 Report

The paper is ok

Author Response

The authors are grateful and would like to thank the reviewer for his for his effort in analyze this work

Round 2

Reviewer 1 Report

Every single comment I had made in my review has been adequately answered. I have also seen the rest of the reviewers and I agree with what they had commented. For my part, the review is adequate, the errors are adequately corrected (both minor and major). I don't see any downside to posting where appropriate.

Author Response

The authors are grateful and would like to thank the reviewer once again

Reviewer 2 Report

In my opinion the article has improved substantially from its initial version to the current one. I think it can be published in its current form

Author Response

The authors thank the reviewer once again for his expertise in this work

Reviewer 3 Report

The quality of the paper has been imporved ad the reviewer comments satisfactorily addressed.

Author Response

(The authors gave the same response as above.)

Reviewer 4 Report

buildings-1118402-peer-review-v2

Dear authors,

The manuscript addresses two important issues, i.e. creep design in concrete and recycled aggregates concrete. According to the text, over 200 studies have been used to determine a new analytical creep model approach for both types of concrete. The notorious dispersion between experimental data and design standards have been clearly set.

But, with all due respect, the following comments should be taken into consideration.

1 The manuscript is full of typos (lines 40 and 41 are empty, capital letter missing in line 154, Table A.1, etc, “à” in Table A.2, the use of “.” instead of “,” in many lines, such as 231, 232, 265, etc.). Also “Figure 7” has been used twice, hence sequential numbering of figures in wrong. All unimportant things.

RV2: Nothing to comment on.

2 According to section 3, the initial approach is based on curves after 90 days of loading. I can assume that this is the best of what the scientific bibliography can offer, but seems insufficient in order to stablished a reformulation of standards. In my opinion, longer-term data are compulsory for this purpose. Moreover, the study of the different parameters involved in the EC design creep formulation is finally based on only two studies, [8] and [14], which, by all means, sounds insufficient in order to propose a modification in the standard. Furthermore, only 60 MPa have been plotted for concretes over 35 MPa.

RV2: Authors agree with the first part; hence we agree the initial approach is inaccurate, especially concerning long-time phenomenon. Of course the study is not limited to two studies, please do not get me wrong, but only two studies were, in my opinion, properly shown as matching data. Please indicate whether references [8] and [41] are shown because they were pioneers on RCA or because these are the best correlated with the model proposed (which in my opinion would be perfectly licit). Please show other studies.

My last comment was about 60 MPa as the only data included in the model for concretes over 35 MPa. What is your consideration about this fact related to the accuracy of your model, taking into account that the aim is to improve an international standard.

3 Paragraph in line 242 assumes insensitivity but at low RH this does not seem correct for both fcm and ho.

RV2: I am sorry, in my opinion, a variation of approximately 30% at RH40% is not negligible and can be reached in service. RH% below 40% data are apparently included in the model, wright? A variation of approximately 25% at RH50% is not negligible and can be reached in service. RH% of 50% data are apparently included in the model, wright? It is my understanding that your affirmation “it is probably not at the origin of the dispersion” should be justified. The expression “probably not” could be considered as unfortunate in a statistical study.

4 Figure 6 shows up to 8 different plots, from a) to g), that should be described in the text. I am sorry but I cannot see the correlation in Figure 6.h. Please explain.

RV2: So is it your understanding that data in figure 6.h are tight? Please show the linear regression line including the formula and R2, and please do it for all plots in Figure 6. I am sorry, I do not see such clear correlation. With all due respect, once more your study is based on relatively little data and inaccurate hypothesis. I am sorry, this is my sincere belief.

5 Please explain how did you modified equation 6 into equation 8 (obviously there is a typo there), but even assuming you are talking about equation 9, its modification only from Figure 6.h should be explained.

RV2: Nothing to comment on.

6 Please explain if formulas in Figure 7 (evolution) are determined by only the few points shown in the figure. Three or four points does not seem sufficient to the purpose.

RV2: Why not showing all points?

7 Paragraph in line 381 indicates that “correlation is better” and that “the number of points does not seem sufficient”, which corroborates the aforementioned, as the figure shows dozens of points. The number of points is insufficient but the correlation, although could be considered as “better” is not good at all.

RV2: You say the number of points is not enough. Then, how do you support your findings?

8 For all the aforementioned, the formulas proposed should not be considered as accurate, even though the improved correlation in Figure 8 (Experimental), as, again, is only based on two studies, hence, generalizations should be avoided.

RV2: Nothing to comment on.

9 The improvement indicated in lines 437-439 is negligible, specially the one shown for EC, which is the proposed for modification. The general modification of a standard should not be based on such small differences.

RV2: We simple have divergent opinions. The variation of R2, considering the limited number of points (not enough as you say), is not sufficient. Furthermore, the variation in mean values is negligible.

10 The improvement shown for RAC is once more based on only two studies.

RV2: I am sorry I did not cover this point adequately, but refer to my RV at point 2.

Author Response

The authors would like to thank the reviewer 4 for his  remarks which led to improve the quality of the manuscript. The detailed answers are given in the attachement
